# Perception of Work-Related Stress and Quality of Life among Nurses during COVID-19 Pandemic–An International, Multicenter Prospective Study

**DOI:** 10.3390/ijerph20043393

**Published:** 2023-02-15

**Authors:** Oľga Jarabicová, Mária Šupínová, Júlia Jankovičová, Izabela Witczak, Miroslava Zemanová, Patrik Christian Cmorej, Łukasz Rypicz

**Affiliations:** 1Faculty of Health Studies, Jan Evangelista Purkyně University in Ústí nad Labem, 40096 Ústí nad Labem, Czech Republic; 2Faculty of Health, Slovak Medical University in Bratislava, 03401 Banská Bystrica, Slovakia; 3Faculty of Nursing and Professional Health Studies, Slovak Medical University in Bratislava, 83101 Bratislava, Slovakia; 4Division of Public Health, Department of Population Health, Faculty of Health Sciences, Wroclaw Medical University, 51-618 Wroclaw, Poland

**Keywords:** nurses, work-related stress, quality of life, pandemic, COVID-19

## Abstract

The major focus of the study was the impact of the COVID-19 pandemic on healthcare workers’ mental health. Nurses are the workers who were exposed to pandemic–related stress, being the most affected. The present cross-sectional study was focused on finding out the differences of the level of work-related stress and quality of life in nurses of the three Central European states, specifically the Czech Republic, the Slovak Republic, and Poland. A structured anonymous online questionnaire was created, and then the link was distributed to the target population through executives. Data analysis was performed by using the R programme–version 4.1.3. The study found that nurses from the Czech Republic achieved lower stress levels and had a higher quality of life than nurses from Poland and Slovakia.

## 1. Introduction

In January 2020, the World Health Organization declared the outbreak of a new coronavirus disease, COVID-19, to be a public health emergency of international concern. COVID-19 affected people from many countries, in many geographical locations. In 2020, countries reported that, on average, about half of the essential health services were disrupted. The COVID-19 pandemic is an example of a fast-emerging pandemic that has placed immense pressure on the healthcare workforce, globally [1].

Nurses play a vital role in providing health services. Nursing professionals constitute approximately half of the global health workforce, and in the current pandemic, they perform the majority of tasks related to preventing and containing infections. Nursing professionals’ role in caring for COVID-19-infected patients and patients’ family members may have negative consequences to their mental health. Increasing COVID-19 cases have coincided with increased workloads, particularly for front-line healthcare professionals. Recognizing potential adverse effects on the mental health of these professionals has instigated research efforts in several countries. For example, the Brasil study by Michele Mandagará de Oliveira et al. (2022) described higher prevalence of depression, anxiety, sleep disorders, and minor psychiatric disorders [2]. Bhandari et al. (2022), in their study, confirmed a similar result: a higher prevalence of depression, anxiety, and stress among 301 nurses in Nepal [3]. The research by Lai et al. (2020) demonstrated the existence of high fear, anxiety, and sleep deprivation in a group of healthcare employees [4]. The publication by Mária Šupínová (2022) and her team showed a worse mental health and quality of life in Slovak nurses [5]. Consequently, suicide rates among nurses are predicted to increase during the COVID-19 pandemic as a result of work-related stressors and mental health problems [6]. The studies of Mónica Blanco–Daza (2022) from Spain and Soon Yeung Bae (2022) from South Korea showed a higher prevalence of post-traumatic stress disorder in nursing staff [7,8]. Nurses play an integral role in pandemic preparedness, yet frequently report to be left out of any response planning or decision making [9,10]. The cross-sectional study of the Czech team of Gurková et al. (2022) dealt with the domains of the work environment, which were significant predictors of missed nursing care activities in Czech hospitals during the COVID-19 pandemic. The higher prevalence of missed nursing care was mainly predicted by the numbers of overtime hours, the nurses’ perception of quality of care, and their satisfaction with their current position [11].

The Polish study of Malinowska-Lipień et al. (2022) examined the relationship between working conditions in Polish hospitals during the COVID-19 pandemic and nurses’ attitudes towards factors promoting patient safety. Working conditions influence nurses’ attitudes towards safety of the hospitalized patients [12].

As can be seen from the above, there are many studies looking at the impact of the COVID-19 pandemic on nurses’ mental health. The present study compares perception of the level of work-related stress (the response people may have when presented with work demands and pressures that are not matched to their knowledge and abilities and which challenge their ability to cope [13]) and quality of life (individual’s perception of their position in life in the context of the culture and value systems in which they live and in relation to their goals, expectations, standards, and concerns [14]) among the Slovak, Czech, and Polish nurses, with an emphasis on the type of care provided in various healthcare facilities during the COVID-19 pandemic. For the purpose of the study, a model was adopted to explain perceptions of work-related stress correlated with quality- of-life dimensions.

The purpose of this study was to compare work-related stress and quality of life among nurses depending on the workplace during the COVID-19 pandemic in three neighboring countries: the Czech Republic, Poland, and Slovakia. Psychosocial strain plays a huge role in the work of nurses, especially in times of huge staff shortages. Knowing the level of work-related stress and quality of life can provide a benchmark for developing preventive programs for this professional group. Hence, conducting research in this area is very important.

## 2. Materials and Methods

### 2.1. Design and Settings

The study was conducted from 1 September 2021 to 30 November 2021. During this time, data were collected. The survey was conducted fully online, using the www.webankieta.pl (accessed on 12 February 2023) platform. For this purpose, 3 electronic versions of the questionnaire were prepared in Polish, Czech, and Slovak languages. Links to the survey were then distributed to nurses working in the healthcare system in Poland, the Czech Republic, and Slovakia. Email addresses of scientific societies dedicated to nurses, social media groups dedicated to nurses, and local government organizations for nurses were used to distribute the link. The survey was fully anonymous and voluntary. To eliminate the risk of duplicate responses by a single person, a database filter was applied, and duplicate IP addresses were discarded. 

### 2.2. Data Collection

Inclusion criteria for the study were possession of a professional nursing license and working in the profession during the COVID-19 pandemic in the healthcare system. A total of 1303 nurses were included in the study: 450 from the Czech Republic, 297 from Poland, and 556 from Slovakia. Additionally, 134 questionnaires were rejected due to incomplete data. The study used random sampling.

### 2.3. Research Tool

The survey instrument consisted of three parts: a socio-demographic metric, questionnaire of Meister, and the SF 36v2. questionnaire.

#### 2.3.1. Questionnaire of the Meister

The study was conducted using a standardized questionnaire of the Meister [15], which determines self-perceived occupational stress. This tool gives information about differences in occupational psychic load perception in many professional groups.

The questionnaire contains ten questions, which are divided into three sections, as follows:(a)overload-time pressure, responsibility, interpersonal conflicts with problems at work;(b)monotony-dissatisfaction, tedious and monotonous work;(c)non-specific load-nervousness, fatigue, exhaustion, dissimilar work performance for longer time.

Respondents expressed their subjective feelings which were assessed on a scale of 1–5:(a)1–no, I totally disagree;(b)2–rather no;(c)3–undecided, sometimes yes, sometimes no;(d)4–rather yes;(e)5–yes, I agree fully.

The results are used for the group assessment. Then, data can be compared with measured critical medians for every question and calculated by the standard method for classification of grade of psychic overload. The load rating has three levels:(a)Psychological stress that is not likely to affect health.(b)Psychological stress, which can periodically affect the subjective state or performance.(c)Psychological stress, in which health risks cannot be ruled out. The assessment of burdening factors is based on exceeding critical median values. However, in our study, we investigated the dependence of the results of the subscales of the Meister questionnaire on the quality of life of nurses.

#### 2.3.2. The SF 36v2. Questionnaire

The study used a shortened version for health-related quality-of-life assessment. The following 11 areas can be assessed:(a)Physical fitness—PF (Physical Functioning)(b)Restriction of activity due to physical problems—RP (Role Physical)(c)Pain—BP (Bodily Pain)(d)General perception of health—GH (General Health)(e)Vitality—VT(f)Social Functioning—SF(g)Sense of mental health—MH (Mental Health)(h)Restriction of activity due to emotional problems—RE (Role Emotional)(i)Change in health—HT (Health Transition)(j)Functioning in the physical dimension, general physical health—PCS(k)Mental functioning, general mental health—MCS

The quality of life in each of the domains is expressed by a number from 0 to 100. Higher numbers mean a better quality of life. 

The authors were licensed to use the SF-36v2 questionnaire [16]—Office of Grants and Research (OGSR) Nonprofit License Agreement, unlicensed: QM056406. The license is necessary because the distributor of the tool is Quality Metric Incorporated.

### 2.4. Ethical Considerations

The study was carried out in accordance with the tenets of the Declaration of Helsinki and guidelines of Good Clinical Practice (World Medical Association, 2013). Written information about the study was provided as an introduction to the survey, with an emphasis on the voluntary and anonymous nature of participation and its guaranteed confidentiality. By answering the questionnaire, participants gave their consent to participate in the study. The research project was approved by the independent Bioethics Committee at the Wroclaw Medical University (No KB–613/2021).

### 2.5. Statistical Analysis

The R program (version 4.1.3.) was used for computations. Chi-squared test (with Yates’ correction for 2 × 2 tables) was used to compare qualitative variables among groups. In case of low values in contingency tables, Fisher’s exact test was used instead. Mann–Whitney test was used to compare quantitative variables between two groups, while Kruskal–Wallis test (followed by Dunn’s post-hoc test) was used for more than two groups. The relationship between two quantitative variables was assessed with Spearman’s coefficient of correlation. The significance level for all statistical tests was set to 0.05.

## 3. Results

### 3.1. The Characteristics of the Study Group

The average age of nurses from all three countries was similar: in the Czech Republic 44.32 ± 11.19, in Poland 45.27 ± 11.11, and in Slovakia 45.31 ± 9.78. No differences were also noted in seniority, as the average obtained for each of the three countries oscillated around 23 years. All groups from each country were heavily feminized, with the percentage of men ranging from 2.36% to 5.33%. The *p*-values of less than 0.05 indicate significant differences between the groups from each country in terms of socio-demographic data, which includes place of residence, as Poles (city over 500th inh. 23.57%) lived in larger cities than Czechs (city over 500th inh. 6.22%) and Slovaks (city over 500th inh. 7.73%) (understandably, Poland has a larger area of the country, which translates into a larger number of large cities). In addition, Poles (71.72%) were more likely to be married than Czechs (58.67%) and Slovaks (59.89%) and less likely to be divorced. In contrast, Slovaks had the most children and Czechs the fewest children. In terms of educational attainment, Poles (14.48%) were more likely to have a master’s degree than Slovaks (12.77%) or Czechs (3.11%). Statistically significant differences were also observed in the parameter of place of work. The most common place of work for Czechs (49.11%) and Poles (55.22%) were hospital departments, while for Slovaks, the place classified as “other” (27.74%). Detailed socio-demographic data can be found in Table 1.

### 3.2. Assessment of Quality of Life by the SF 36 Questionnaire

This study also analyzed the quality of life of nurses from the Czech Republic, Poland, and Slovakia. The quality-of-life results were presented in 11 dimensions (detailed results are summarized in Table 2). Values of *p* < 0.05 indicate the following statistically significant relationships:(a)Quality of life in the domains of PF (Physical Functioning), BP (Bodily Pain), MH (Mental Health), and HT (Health Transition) was significantly higher in Czechs than in Poles and Slovaks.(b)Quality of life in the domains of PF (Physical Functioning), VT (Vitality), RP (Role Physical), and RE (Role Emotional) was significantly higher in Czechs and Poles than in Slovaks.(c)Quality of life in the domains of PF (Physical Functioning), GH (General Health), SF (Social Functioning), PCS (Functioning in the physical dimension, general physical health), and MCS (Mental functioning, general mental health) was significantly higher in Czechs than in Poles and Slovaks, and, moreover, was significantly higher in Poles than in Slovaks.

It should be noted that the differences obtained in the study population in the three countries are statistically significant. Nevertheless, the obtained values in only a few dimensions are noticeably higher/lower (e.g., BP, GH, and HT). This may be compounded by the cultural proximity of the countries involved in the study, the competence of the nurses, and the comparable lifestyles of the population, and thus the nurses. It should also be noted that in areas where Poland directly borders the Czech Republic, cultures or lifestyles intermingle. Nurses from Poland work in Czech hospitals and vice versa.

### 3.3. Assessment of Work-Related Stress by the Meister Questionnaire

The level of work-related stress as a whole was evaluated (detailed results are summarized in Table 3 and Table 4). The study showed statistically significant correlations (*p* < 0.05):(a)The level of overload was significantly higher in Poles than in Czechs and Slovaks, and was also significantly higher in Slovaks than in Czechs;(b)The level of monotony, non-specific factor, and overall level of burden were significantly higher in Poles and Slovaks than in Czechs.

Indeed, the nursing staff in Poland is more overloaded with work. This can be referred to the values of the ratios of the number of nurses per 1000 patients in which Poland has the lowest. The reason for this can be found not only in the shortage of nursing staff, but because the organization of healthcare and work itself in all three countries is very similar, cultural differences are virtually unnoticed when it comes to the size of work.

## 4. Discussion

In this study, the authors attempted to compare the level of work-related stress and quality of life in nursing staff from the Czech Republic, Slovakia, and Poland during the COVID-19 pandemic. This international, multicenter study makes it possible to determine the relationship between the workplace and stress and quality of life in three European neighboring countries. It should be noted that the timing of the study included a particularly difficult period for medical personnel, namely the COVID-19 pandemic, which should be considered an additional stress factor affecting both stress levels and quality of life. 

A shortage of nursing staff can be seen in all the countries surveyed, but the Czech Republic compares much better with Poland and Slovakia. According to the report Health at a Glance 2021: OECD Indicators [17], the Czech Republic has 8.7 nurses per 1000 inhabitants, Slovakia 5.8/1000, and Poland 5.1/1000. The average value of the number of nurses per 1000 inhabitants for OECD countries was 8.8. During the pandemic, there was a very high demand for nursing staff virtually all over the world. The age of nurses is also not insignificant and can affect stress levels or quality of life. In the Czech Republic, the average age of nurses is 46.1 years (nurses with specialization) [18], in Slovakia in 2019, it was 46.9 years [19], and in Poland in 2020, it was 53.2 years [20]. 

The results obtained in the study clearly indicate that lower levels of stress—regardless of its dimension—translate into higher quality of life for nurses. Czech nurses have lower levels of stress than nurses from Poland or Slovakia. This translates directly into their quality of life, which is also higher compared to nurses from the neighboring countries. A similar relationship between stress levels and quality of life in nurses in their studies was shown by Tabrizi et al. [21], Amjadi et al. [22], and Babapour et al. [23].

The nurses from the three countries studied were of similar ages and had similar work experience. What, then, could be the reason that nurses from the Czech Republic achieve lower levels of stress and higher levels of quality of life? It can be surmised that the underlying cause is excessive workload. If we look at the ratio of nurses per 1000 patients, we can see that it reaches the highest value in the Czech Republic. Nursing shortages may manifest themselves in this way. In addition, it may be related to care rationing, since the need to ration care—resulting, among other things, from staff shortages—affects the quality of life [24,25]. An interesting relationship in the values obtained for stress level and quality of life can be seen in Polish and Slovak nurses. Similar results were obtained for both countries. If we take into account that the values of the index of the number of nurses per 1000 patients in Poland and Slovakia are similar, we can see the reason in this aspect.

Interestingly, similar results were obtained in a study conducted by Debska et al. (2018), which also compared the performance of nurses from Poland, the Czech Republic, and Slovakia in terms of job burnout and work-related stress [26]. The study was implemented before the pandemic and concerned anesthesia nurses. A group of nurses from the Czech Republic achieved scores on subscales of work-induced stress that were statistically significantly different from nurses from Poland and Slovakia. In contrast, no statistically significant differences were observed in the group of Polish vs. Slovak nurses.When analyzing the results, the aspect related to the pandemic cannot be overlooked. The appearance in the world of a new variant of the coronavirus-SARS-CoV-2 has caused total disorganization and paralysis of the healthcare systems around the globe. Fear of contracting the new virus, fear of transmitting the virus to one’s loved ones, fear of death due to SARS-CoV-2 virus infection, and social exclusion are just a few examples of stress factors that affected nursing staff during the pandemic, which also affected their quality of life [27,28,29,30]. 

## 5. Conclusions

The work of nurses is burdened with many stressors, which can directly or indirectly affect a person’s perceived stress level. Stress, in turn, affects the quality of life, and the transfer of problems from work to the private sphere is a common phenomenon. This study showed that the level of stress affects the quality of life of nurses. It is worth looking at the factors that cause stress in the work of nurses. Undoubtedly, it can be staffing shortages that cause excessive workloads and can induce rationing of care and many others. Healthcare systems are facing a staffing deficit that will not be solved overnight. Therefore, it is very important to provide nurses with preventive measures. The ergonomics of the workplace, including psychosocial factors, are very important. At present, the implementation of preventive programs to reduce stress levels among nursing staff may be the only effective action in the field of professional work against the reduction of quality of life. As part of improving the ergonomics of the workplace and to avoid monotony, which can contribute to significant overload, mandatory rotations within different hospital departments can be implemented (e.g., from an oncology department with a very high psychosocial load to an internal medicine department). In addition, some hospitals are already implementing work–life balance programs and group classes with a psychologist to reduce bad emotions. These are preventive measures that can be implemented in any facility.

### Study Limitations

The authors recognize some limitations of the study. Assuming random selection of study participants, it is difficult to assess whether there were any limitations in access to the entire geographical range of participants. Since the survey was conducted online, people who are not so eager to use the Internet could have been excluded—this also includes digital exclusion.

## Figures and Tables

**Table 1 ijerph-20-03393-t001:** Characteristics of the studied group of nurses.

Parameter		Country			*p*
		Czechia (N = 450)	Poland (N = 297)	Slovakia (N = 556)	
Age [years]	mean ± SD	44.32 ± 11.19	45.27 ± 11.11	45.31 ± 9.78	*p* = 0.196
	median	46	48	46	
	quartiles	38–52	37–54	41–52	
Seniority [years]	mean ± SD	23.17 ± 12.11	23.01 ± 12.56	23.77 ± 11.54	*p* = 0.675
	median	25	27	25	
	quartiles	15–32	11–33	15–33	
Sex	Female	426 (94.67%)	290 (97.64%)	536 (96.40%)	*p* = 0.107
	Male	24 (5.33%)	7 (2.36%)	20 (3.60%)	
Residence	Rural area	148 (32.89%)	116 (39.06%)	233 (41.91%)	*p* < 0.001 *
	City up to 50th inh.	112 (24.89%)	61 (20.54%)	150 (26.98%)	
	City 50–100th inh.	116 (25.78%)	15 (5.05%)	81 (14.57%)	
	City 100–500th inh.	46 (10.22%)	35 (11.78%)	49 (8.81%)	
	City over 500th inh.	28 (6.22%)	70 (23.57%)	43 (7.73%)	
Marital status	Single	52 (11.56%)	35 (11.78%)	77 (13.85%)	*p* < 0.001 *
	Married	264 (58.67%)	213 (71.72%)	333 (59.89%)	
	Divorced	59 (13.11%)	13 (4.38%)	66 (11.87%)	
	Informal relationship	61 (13.56%)	28 (9.43%)	50 (8.99%)	
	Single parent	6 (1.33%)	3 (1.01%)	12 (2.16%)	
	Widow(er)	8 (1.78%)	5 (1.68%)	18 (3.24%)	
No of children	No children	95 (21.11%)	70 (23.57%)	135 (24.28%)	*p* = 0.003 *
	1 or 2 children	317 (70.44%)	189 (63.64%)	333 (59.89%)	
	3 or more children	38 (8.44%)	38 (12.79%)	88 (15.83%)	
Education	high school without specialization	0 (0.00%)	27 (9.09%)	0 (0.00%)	*p* < 0.001 *
	high school and specialization	295 (65.56%)	29 (9.76%)	199 (35.79%)	
	bachelor without specialization	64 (14.22%)	60 (20.20%)	101 (18.17%)	
	bachelor and specialization	36 (8.00%)	40 (13.47%)	32 (5.76%)	
	master without specialization	14 (3.11%)	43 (14.48%)	71 (12.77%)	
	master and specialization	37 (8.22%)	92 (30.98%)	146 (26.26%)	
	doctoral without specialization	4 (0.89%)	1 (0.34%)	7 (1.26%)	
	doctoral and specialization	0 (0.00%)	5 (1.68%)	0 (0.00%)	
Workplace	Primary care	9 (2.00%)	35 (11.78%)	52 (9.35%)	*p* < 0.001 *
	Ambulatory healthcare	47 (10.44%)	9 (3.03%)	75 (13.49%)	
	Emergency department	23 (5.11%)	9 (3.03%)	10 (1.80%)	
	Intensive care unit/anesthesiology department	75 (16.67%)	20 (6.73%)	98 (17.63%)	
	Surgical ward or non-surgical hospital ward	221 (49.11%)	164 (55.22%)	117 (21.04%)	
	Hospital department for patients with COVID-19	8 (1.78%)	4 (1.35%)	53 (9.53%)	
	Temporary hospital for patients from COVID-19	2 (0.44%)	6 (2.02%)	5 (0.90%)	
	SARS-CoV-2 Test Point	4 (0.89%)	0 (0.00%)	8 (1.44%)	
	COVID-19 vaccination centre	6 (1.33%)	4 (1.35%)	6 (1.08%)	
	Others	55 (12.22%)	46 (15.49%)	132 (23.74%)	

*p*–Kruskal–Wallis test for quantitative variables, chi-squared or Fisher’s exact test for qualitative variables. * Statistically significant (*p* < 0.05).

**Table 2 ijerph-20-03393-t002:** A comparison of the quality of life of nurses from the Czech Republic, Poland, and Slovakia.

SF36v2		Country			*p*
		Czechia (N = 450)	Poland (N = 297)	Slovakia (N = 556)	
PF	mean ± SD	86.48 ± 14.97	80.62 ± 19.68	79.38 ± 18.39	*p* < 0.001 *
	median	90	85	85	
	quartiles	80–100	75–95	70–95	CZ > PL, SK
RP	mean ± SD	68.04 ± 20.88	68.12 ± 20.87	60.85 ± 22.07	*p* < 0.001 *
	median	68.75	68.75	62.5	
	quartiles	51.56–81.25	50–87.5	50–75	PL, CZ > SK
BP	mean ± SD	73.26 ± 23.62	58.4 ± 24	58.35 ± 23.35	*p* < 0.001 *
	median	77.78	55.56	55.56	
	quartiles	55.56–100	44.44–77.78	44.44–77.78	CZ > PL, SK
GH	mean ± SD	67.43 ± 19.25	58.96 ± 17.51	54.42 ± 19.55	*p* < 0.001 *
	median	70	60	55	
	quartiles	55–83.75	50–70	40–70	CZ > PL, SK PL > SK
VT	mean ± SD	50.89 ± 18.89	48.67 ± 20.19	40.65 ± 20.1	*p* < 0.001 *
	median	50	50	37.5	
	quartiles	37.5–62.5	37.5–62.5	25–56.25	CZ, PL > SK
SF	mean ± SD	69.11 ± 24.44	56.19 ± 26.36	51.19 ± 24.91	*p* < 0.001 *
	median	75	50	50	
	quartiles	50–87.5	37.5–75	37.5–62.5	CZ > PL, SK PL > SK
RE	mean ± SD	70.81 ± 22.1	71.13 ± 22.58	66.56 ± 23.48	*p* = 0.005 *
	median	75	75	66.67	
	quartiles	50–91.67	58.33–91.67	50–83.33	PL, CZ > SK
MH	mean ± SD	63.96 ± 17.34	56.55 ± 18.36	56.41 ± 19.09	*p* < 0.001 *
	median	65	60	55	
	quartiles	50–75	45–70	45–70	CZ > PL, SK
HT	mean ± SD	41.11 ± 16.05	36.03 ± 17.99	34.76 ± 21.02	*p* < 0.001 *
	median	50	50	25	
	quartiles	25–50	25–50	25–50	CZ > PL, SK
PCS	mean ± SD	74.25 ± 15.49	67.8 ± 16.42	64.23 ± 17.11	*p* < 0.001 *
	median	76.92	67.69	64.62	
	quartiles	63.46–86.15	56.92–80	52.31–76.92	CZ > PL, SK PL > SK
MCS	mean ± SD	62.43 ± 17.19	57.37 ± 18.8	53.34 ± 18.4	*p* < 0.001 *
	median	63.39	57.14	53.57	
	quartiles	50–75	42.86–73.21	39.29–66.07	CZ > PL, SK PL > SK

*p*–Kruskal–Wallis test + post-hoc analysis (Dunn’s test). * statistically significant (*p* < 0.05).

**Table 3 ijerph-20-03393-t003:** Group assessment of Meister questionnaire in nurses by country.

HPZ		Country			*p*
		Czechia (N = 450)	Poland (N = 297)	Slovakia (N = 556)	
Overload	mean ± SD	8.24 ± 2.41	10.78 ± 2.34	9.66 ± 2.67	*p* < 0.001 *
	median	8	11	10	
	quartiles	6–10	9–12	8–12	SK > CZ PL > SK, CZ
Monotony	mean ± SD	5.02 ± 2.03	6.05 ± 2.42	6.41 ± 2.64	*p* < 0.001 *
	median	4.5	6	6	
	quartiles	3–6	4–7	4–8	SK, PL > CZ
Non-specific load	mean ± SD	9.29 ± 3.71	11.79 ± 4.64	12.22 ± 4.13	*p* < 0.001 *
	median	9	12	12	
	quartiles	6–12	8–15	9–15	SK, PL > CZ
Total load	mean ± SD	22.56 ± 6.94	28.62 ± 8.22	28.29 ± 8.27	*p* < 0.001 *
	median	22	28	29	
	quartiles	17–27	22–35	22.75–34	SK, PL > CZ

*p*–Kruskal–Wallis test + post-hoc analysis (Dunn’s test). * statistically significant (*p* < 0.05).

**Table 4 ijerph-20-03393-t004:** Assessment of critical medians in nurses by country.

Nurses (N = 1303)	Czech (N = 450)	Poland (N = 297)	Slovakia (N = 556)	Critical Value of Medians	*p* Value	Post-Hoc
FACTOR I. (OVERLOAD) Time pressure Responsibility Interpersonal conflicts	3.5 3 2	4 5 2	4 4 2	3 3 2.5	*p* < 0.001 * *p* < 0.001 * *p* < 0.001 *	PL, SK > CZ PL > SK > CZ PL, SK > CZ
FACTOR II. (MONOTONY) Dissatisfaction Tedious work Monotony	2 1 1	2 2 2	3 2 2	2.5 2.5 2.5	*p* < 0.001 * *p* < 0.001 * *p* < 0.001 *	SK, PL > CZ SK > PL, CZ SK, PL > CZ
FACTOR III. (NON SPECIFIC LOAD) Nervousness Exhaustion Fatigue Decrease of long work performance	2 2 2 2	3 2 3 3	3 3 3 4	3 3 3 2.5	*p* < 0.001 * *p* < 0.001 * *p* < 0.001 * *p* < 0.001 *	PL > SK > CZ SK > PL > CZ PL, SK > CZ SK > PL > CZ

* statistically significant (*p* < 0.05).

## Data Availability

Not applicable.

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
