# Peer review of "Perception of Work-Related Stress and Quality of Life among Nurses during COVID-19 Pandemic–An International, Multicenter Prospective Study"

_ijerph, 2023, doi:10.3390/ijerph20043393_

Round 1

Reviewer 1 Report

Brief summary

By the way of an online questionnaire the authors compared socio-demographic data, work related stress and quality of life among nurses during Covid-19 pandemic, between three central European countries: Czech Republic, Slovak Republic and Poland. They found differences between these countries that are mainly related to the shortage of nursing staff in these countries. Shortage induces work load, time pressure dissatisfaction, exhaustion, etc. Consequently, it is not surprising that the country with the higher staffing rate (number of nurses per 1’000 inhabitants) has the better results. This is the Czech Republic.

General comments

The idea to investigate work-related stress and quality of life among nurses during the Covid-19 pandemic was very good since this population has been exposed to a high level of stress including the fear of contracting the virus with possible grave consequences on health. The suffering of this target population has been a major occupational health and public health problem. However, the strategy chosen by the authors has been focused exclusively on the individual perception of stress and quality of life without any consideration on the working environment: work organization, leadership, management, governance, collaboration, etc. [1]. Therefore, the results of this study are just a statement of a situation without any analysis of the roots of the problems and the way they could have been prevented or reduced. Even if the way this research has been conducted is correct it gives an incomplete picture of the situation and is therefore disappointing.

Specific comments

Questionnaires: Three questionnaires have been used for this survey but no reference is given to allow the reader to look further to these tools. It should also be explained why the users of the SF 36v2 questionnaire must be licensed.

Characteristics of the studied group of nurses: this socio-demographic survey looks exhaustive and gives an apparently reliable picture of the three population of nurses. From a sociology point of view, it is interesting.

Questionnaire of the Meister: this tool to register the self-perceived occupational stress sounds well-structured and enough detailed to get an interesting description of the work-related stress of these nurses.

Questionnaire SF 36v2.: the assessment of the quality of life by this instrument is much less convincing than the other ones. The reasoning associated to the selection of the 11 areas is not clear and looks arbitrary. It appears as a blurred mixture of effects or perceptions related to physical and mental health, forgetting the social and moral dimensions. Therefore, the results are difficult to interpret and the differences between countries do not bring relevant information.

Results from the SF 36v2. Questionnaire (Chapter 3.2)

The differences between countries are simply stated with no suggestions about their origin. So, they did not bring any interesting elements.

Results from the Meister questionnaire (Chapter 3.3 which is erroneously written 3.2)

The same comment applies to these results.

Conclusions (Chapter 5)

At the end of this chapter, it is mentioned that it is very important to provide nurses with preventive measures in the area of ergonomics and psychosocial factors. But this remains a wishful thinking without any clear suggestions issued from the results of this study.

Study Limitations (End of chapter 5)

The authors stress that the study lacks a control group but they do not cite an interesting study made in 2108 in the three concerned countries on the psychosocial burden and occupational burnout of nurses investigated with the questionnaire of Meister that could allow interesting comparisons [2]. Is it an oversight?

Conclusion of this review

All the comments mentioned above lead to the conclusion that  this study (made during a period that is now over) does not bring enough relevant results to be really prospective and constructive.

The differences shown between countries should be interpreted with some assumptions about culture, economic and political factors, etc.

Moreover, correlations between and within factors (socio-demographic, work-stress and quality of life) should be investigated (analyzed) also with and between countries. Probably interesting elements may emerge giving much more value to this study and allowing a better perception the influence of preventive measures may have on other factors. 

References

[1] Mandawala, D.E.N. Karunanayake P, (2021) A study of the effect of selected organizational factors on perceived stress of nurses : Case study of a leading private hospital in Sri Lanka. Academia Letters Article 360. https://doi.org/10.20935/AL3670

[2] Debska, G. Haducàkovà, H. Kràtkà, A. and Pasek, M. (2019) Assessment of Psychosocial Burden and Occupationl Burnout in Nurses Working in Intensive Care Units in Poland, Slovakia and the Csech Republic. Clinical Social Work and Health Intervention 10(2): 53-61 DOI:10.22359/cswhi_10_2_08

Author Response

Dear Reviewer,
Thank you for your time. I believe that all addressed comments will make the manuscript fit for publication.

Łukasz Rypicz

Reviewer 2 Report

The work addresses the current issue of analysing the possible consequences of COVID-19 in one of the professional groups that was most directly affected by the pandemic, nursing, although I do not clearly see the possible interest of the results if possible interventions to reduce these effects are not presented on the basis of the results.

In my opinion, in order for this research to be published, the following aspects should be modified:

1. Perceived stress should appear in the title.

2. Better justify the relevance of the study in the introduction, why and for what purpose the study is being carried out.

3. The introduction should indicate how the authors understand stress and quality of life, as well as under which models they study these variables.

4. The method section would be much clearer, in my opinion, if it were developed following the classic sub-sections of the method: participants, instruments and procedure.

5. It should specify whether the sample was representative.

6. It does not clarify the type of sampling used to establish the sample.

7. The procedure should be more clearly detailed so that it can be replicated by another researcher who wishes to do so.

8. I agree with the limitation presented by the authors, however, I consider that more limitations can be included, depending on the study carried out.

Author Response

(The authors gave the same response as above.)

Round 2

Reviewer 1 Report

Authors' responses are satisfactory and the changes and additions made to the text have improved its quality.

I remain highly sceptical about the questionnaire SF 36v2 but I can understand the authors' arguments.